# Obesity and Cancer Rehabilitation for Functional Recovery and Quality of Life in Breast Cancer Survivors: A Comprehensive Review

**DOI:** 10.3390/cancers16030521

**Published:** 2024-01-25

**Authors:** Lorenzo Lippi, Alessandro de Sire, Arianna Folli, Alessio Turco, Stefano Moalli, Marco Marcasciano, Antonio Ammendolia, Marco Invernizzi

**Affiliations:** 1Department of Health Sciences, University of Eastern Piedmont “A. Avogadro”, 28100 Novara, Italy; lorenzo.lippi@ospedale.al.it (L.L.); 20042370@studenti.uniupo.it (A.F.); alessio.turco.phys@gmail.com (A.T.); stefano.moalli@libero.it (S.M.); marco.invernizzi@med.uniupo.it (M.I.); 2Translational Medicine, Dipartimento Attività Integrate Ricerca e Innovazione (DAIRI), Azienda Ospedaliera SS. Antonio e Biagio e Cesare Arrigo, 15121 Alessandria, Italy; 3Department of Medical and Surgical Sciences, University of Catanzaro “Magna Graecia”, 88100 Catanzaro, Italy; ammendolia@unicz.it; 4Research Center on Musculoskeletal Health, MusculoSkeletalHealth@UMG, University of Catanzaro “Magna Graecia”, 88100 Catanzaro, Italy; 5Experimental and Clinical Medicine Department, Division of Plastic and Reconstructive Surgery, University of Catanzaro “Magna Graecia”, 88100 Catanzaro, Italy; m.marcasciano@unicz.it

**Keywords:** breast cancer, morbid obesity, obesity management, rehabilitation, personalized medicine, physical exercise, health-related quality of life, treatment outcome

## Abstract

**Simple Summary:**

Obesity is a global health challenge closely linked to breast cancer (BC). Recognizing this intricate relationship, our research explores the multifaceted connections between obesity and BC. The aim of this narrative review was to underline the key role of a personalized rehabilitation approach targeting the connection between obesity and BC. By comprehending these links, we aim to enhance risk assessment, improve survivorship care, and contribute valuable insights to the broader field of cancer rehabilitation. Moreover, tailored and technology-driven methods show potential insight into enhancing the effectiveness of rehabilitation in cancer patients with obesity. Through this research, we seek to pave the way for effective strategies addressing the challenges faced by obese individuals with BC.

**Abstract:**

Obesity is a global health challenge with increasing prevalence, and its intricate relationship with cancer has become a critical concern in cancer care. As a result, understanding the multifactorial connections between obesity and breast cancer is imperative for risk stratification, tailored screening, and rehabilitation treatment planning to address long-term survivorship issues. The review follows the SANRA quality criteria and includes an extensive literature search conducted in PubMed/Medline, Web of Science, and Scopus. The biological basis linking obesity and cancer involves complex interactions in adipose tissue and the tumor microenvironment. Various mechanisms, such as hormonal alterations, chronic inflammation, immune system modulation, and mitochondrial dysfunction, contribute to cancer development. The review underlines the importance of comprehensive oncologic rehabilitation, including physical, psychological, and nutritional aspects. Cancer rehabilitation plays a crucial role in managing obesity-related symptoms, offering interventions for physical impairments, pain management, and lymphatic disorders, and improving both physical and psychological well-being. Personalized and technology-driven approaches hold promise for optimizing rehabilitation effectiveness and improving long-term outcomes for obese cancer patients. The comprehensive insights provided in this review contribute to the evolving landscape of cancer care, emphasizing the importance of tailored rehabilitation in optimizing the well-being of obese cancer patients.

## 1. Introduction

Obesity is currently considered a global health challenge with a correlation of several pathological conditions and consequently with a multifaceted impact overall well-being [1,2]. Moreover, it is predicted that its related disability-adjusted life years (DALYs) and deaths trends will follow an upward trajectory within the next years, also considering the potential negative influence of bariatric surgery in the most severe cases [3,4]. In this context, a growing body of evidence has highlighted the intricate relationship between obesity and cancer. This linkage, once regarded as an emerging concern, has now turned into a critical issue in both oncology and public health [5]. It is clear that with the increasing prevalence of obesity worldwide [1], the impact of this linkage on cancer incidence and outcomes cannot be overstated.

The connection between obesity and cancer is multifactorial [6], encompassing alterations in hormonal profiles, chronic low-grade inflammation, and the intricate interplay of various metabolic pathways. This nexus is not confined to a single type of cancer; instead, it spans a broad spectrum, including but not limited to colorectal [7], pancreatic [8], liver [9], endometrial, ovarian, adenocarcinoma of the esophagus, gastric cardia, gallbladder, renal cell carcinoma, thyroid cancers, as well as meningioma and multiple myeloma [10] and, last but not least, postmenopausal breast cancer (BC) [10,11].

In 2018, with 2.1 million new cases diagnosed, BC represented the most frequently detected cancer in the female gender. Additionally, it was the primary cause of cancer-related fatalities in women worldwide, leading to 627,000 deaths in the same year [10]. On the other hand, mortality in first world countries has decreased, such as Australia, Canada, and the USA, due to the increasing effectiveness of screening programs and treatment protocols [10,12]. Concurrently to the increasing prevalence of cancer survivors, attention is rising on rehabilitation interventions enhancing overall well-being before, during, and after treatment, aiming at restoring or improve quality of life [12].

In this context, understanding the obesity–cancer linkage is crucial in the precise management of cancer patients, promoting a specific risk stratification and enabling the identification of individuals at higher risk with a targeted screening [11]. Moreover, obesity has important treatment implications, since it can affect treatment responses and the side effect profile of cancer therapies, necessitating tailored treatment plans for obese patients [13,14]. Additionally, addressing the obesity–cancer linkage is critical for cancer survivors, as obesity can exacerbate the risk of cancer recurrence and complicate the management of long-term side effects [15].

In this scenario, growing research is now focusing on the rehabilitation management of patients with breast cancer, aiming at enhancing functional recovery and optimizing HR-QoL. Interestingly, BC rehabilitation encompasses a holistic approach to patient care, emphasizing physical, psychological, and nutritional well-being [16,17,18,19]. In cancer rehabilitation, the management of body weight has a crucial effect on functional recovery since obesity leads to poorer cancer prognosis, poorer surgical outcomes, higher risk of lymphedema, fatigue, and a decline in overall quality of life [12,20,21]. Despite these considerations, evidence supporting the role of obesity in cancer rehabilitation is inconclusive and still evolving. While several studies suggest a link between obesity and poorer treatment outcomes, the specific impact of obesity on rehabilitation interventions remains a complex area requiring further investigation.

In light of these considerations, the goal of this narrative review is to outline the current evidence regarding the association between obesity, cancer, and rehabilitation. The aim is to provide a more in-depth understanding and insight into the most effective rehabilitation intervention for individuals with both obesity and breast cancer.

## 2. Materials and Methods

This narrative review was conducted in accordance with the established quality criteria outlined in the SANRA framework [22]. The SANRA score were reported in the Appendix A. Comprehensive literature searches were performed across multiple databases, including PubMed/Medline, Web of Science (WoS), and Scopus, utilizing controlled vocabulary terms such as “Obesity”, “Cancer”, “Rehabilitation”, “Exercise”, “Nutrition”, “Rehabilitation”, “Physical Activity”, “Lifestyle”, “Function”, “Body Composition”, “Pain”, “Performance”, “Disability”, “Quality of Life”, “Cancer Survivorship”. The search strategy, following the SPIDER tool approach [23], is summarized in Table 1.

Between June 2023 and November 2023, two independent reviewers (L.L. and A.F.) carried out the literature search. Subsequently, the identified studies underwent a rigorous eligibility screening process conducted by the reviewers. In cases where consensus could not be reached, a third reviewer (A.d.S.) was consulted to make a final judgment. The inclusion criteria for this review encompassed studies addressing the primary research question: “What are the most effective rehabilitation strategies for addressing the multifaceted challenges faced by obese patients with BC?” Specifically, eligible articles focused on human subjects diagnosed with BC and obesity, explored the multidimensional aspects of disability in this patient population, and assessed the impact of rehabilitation interventions on improving functional outcomes, quality of life, and psychosocial well-being. Studies in languages other than English, those lacking full-text availability, those not involving human subjects, and conference abstracts or theses were excluded.

Data extraction and synthesis were carried out using qualitative methods. The reviewers (L.L. and A.F.) independently extracted and synthesized information on rehabilitation strategies tailored for obese patients with cancer. In cases of discordance, input from a third reviewer (M.I.) was sought to reach a consensus. Given the heterogeneity of the included studies and the narrative review design, a qualitative synthesis approach was employed, presenting all outcome data in a narrative format. The reliance on qualitative research pertains to the type of data synthesis for the review, not the types of studies considered.

## 3. Biological Basis Linking Obesity and Cancer

Adipose tissue is a complex organ composed of adipocytes, mesenchymal stromal cells, macrophages, dendritic cells, mast cells, eosinophils, neutrophils, and T and B lymphocytes [24]. Within its microenvironment, adipocytes produce bioactive molecules, factors that collectively could contribute to tumor behavior [25,26].

In this context, growing attention is rising on adipose tissue endocrine functions, characterized by the secretion of adipokines and hormones with increasing production and circulating levels with the increasing of adipose tissue mass [27,28]. Among the most important are leptin and adiponectin (which, respectively, increase and decrease in obesity), tumor necrosis factor (TNF) α, interleukin (IL)-6, and insulin-like growth factor-1 (IGF-1), CXCL12, steroid hormones, and lipids. These factors can promote cancer growth through oncogenic signaling or through indirect pathways, such as angiogenesis and immunomodulation [24].

On the other hand, the tumor-specific microenvironment is a complex set of cells and molecules surrounding tumors. This system includes tumor endothelial cells, tumor stromal cells with cancer-associated fibroblasts and cancer-associated adipocytes, normal epithelial cells, immune cells with tumor-associated macrophages, signaling molecules, blood vessels, and a non-cellular part called the extracellular matrix [29]. Leptin and adiponectin play intricate roles within the tumor microenvironment. More specifically, leptin can directly and indirectly impact on inflammation and angiogenesis, while adiponectin’s role is not fully understood but appears to be related to antioxidant markers and tumor outcomes in different cancers. These adipokines contribute to the complex interplay of factors within the tumor microenvironment, potentially influencing cancer progression [29].

In addition, the presence of aromatase in peripheral adipose tissue can lead to an increased conversion of androgens into estradiol. This elevated production of estrogen by adipose tissue has been associated with an increased risk of developing breast, endometrial, ovarian, and other types of cancers [6,28]. In other words, in postmenopausal women, obesity might implicate an increase in breast cancer development: in fact, the accumulation of additional fat tissue in women can increase breast cancer incidence due to the increase in estrogen and insulin levels [28,30]. Figure 1 shows in detail the adipose tissue impact on tumor environment.

Furthermore, in obese individuals, there is often an elevation in insulin and IGF-1 levels in the bloodstream. Insulin resistance, a known cancer risk factor, leads to high insulin levels, known as hyperinsulinemia, occurring before the onset of type 2 diabetes. Elevated insulin and IGF-1 levels are believed to play a role in the development of cancers by promoting cellular proliferation and inhibiting apoptosis—potentially allowing tumor growth [31], such as colon, renal, prostate, and endometrial cancer [6,31]. Inflammation, in this context, is not merely a bystander but an active participant in the cancer cascade [32].

Moreover, the presence of immune system cells can play a role in inflammation of adipose tissue: in fact, after reaching a certain overgrowth, the tissue becomes inflamed and fibrotic, developing a chronic inflammation that has implications for tumorigenesis [24]. In this context, chronic inflammation can be considered a hallmark of obesity, and as a possible player. The sustained release of proinflammatory cytokines such as IL-6 and TNF-α from adipose tissue creates an environment that favors carcinogenesis [32].

Another proposed mechanism involves the CXCL12/CXCR4 pathway. CXCR4 plays a pivotal role as chemokine receptor in cancer cells, representing the most common type and CXCL12 is its ligand [33]. CXCR4 is present in but not limited to kidney, lung, brain, prostate, and breast cancers. In animal studies, CXCL12 was found to be secreted by adipose tissue and played a role in insulin resistance obesity correlated and adipose tissue inflammation [5].

It should be noted that an increase in pro-inflammatory cytokines leads to increased production of reactive oxygen species (ROS) and nitrogen by inflammatory cells, this could be again correlated as an increased risk due to an increased presence of adipose tissue, leading to a rise in oxidative stress (OS) [34]. In turn, by a vicious cycle, OS promotes low-grade chronic inflammation in adipose tissue [34]. At the same time, ROS and antioxidant transcription factors might play a role in cancer development and progression [35]. ROS can have both stimulating and cytotoxic effects on cancer cells, and tumor cells adapt to high ROS levels by modifying their metabolism and activating antioxidant genes [35]. The interplay between ROS and antioxidant systems in cancer development is complex and a potential target in cancer treatment, though further research is needed [35].

In addition, mitochondrial dysfunction might have potential implications for obesity and cancer linking [36]. In addition, the mitochondria have a central role in energy metabolism; they are also involved in the production and elimination of ROS, with the mitochondrial respiratory chain being the main source of ROS during adenosine triphosphate (ATP) production. When nutrient signal input, energy production, and oxidative respiration become imbalanced, it results in mitochondrial dysfunction, which encompasses various metabolic perturbations, eventually leading to mitochondrial dysfunction. Obesity has been linked to mitochondrial dysfunction, associated with metabolic changes, insulin resistance, and inflammation, with potential implications for the development and progression of obesity-related complications [36]. Abnormal mitochondrial dynamics have also been linked to malignant transformation and the development of cancer [37]. The loss of proper mitochondrial dynamics can have a significant impact on cell growth and survival [37].

Lastly, obesity has also been linked to shorter telomeres, likely due to oxidative stress and inflammation, potentially contributing to cancer [38,39]. Telomere length is crucial for the full replication of DNA and plays a vital role in maintaining the stability and integrity of chromosomes. Telomere length naturally decreases with each subsequent mitotic cell division, and as it shortens, it can either impede cell growth or trigger cell death through a signal indicating cell replication aging [38]. Telomere length is a critical factor in cancer development. Paradoxically, both short and long telomeres are associated with cancer risk. Telomerase, an enzyme responsible for maintaining telomere length, is frequently expressed in tumors, allowing for unlimited cell growth [39]. Telomere dysfunction, caused by critical shortening, can either suppress or promote cancer, depending on the cellular context. Shortened telomeres may trigger cell cycle arrest or apoptosis, but in some cases, cells bypass these signals, leading to malignant transformation [39].

Taken together, the accumulation of excess body fat is related to an increase of 17% risk of cancer-specific mortality [6], but also to an increased risk of all-cause mortality [40]; the risk of cancer death is higher in patients in patients suffering from grade III obesity [40]. Though the association between obesity and cancer is far from being understood, its linkage might represent a crucial target for therapeutic interventions [6,41,42,43]. Trials exploring the impact of weight management strategies and lifestyle modifications on cancer risk and outcomes suggest that intervening in the obesity–cancer linkage through weight loss interventions may hold promise for reducing cancer risk and improving outcomes in those with obesity-related malignancies [44,45].

Epidemiological evidence plays another pivotal role in clarifying the intricate relationship between obesity and cancer at the population level [46]. Numerous epidemiological studies have consistently demonstrated associations between excess body weight and an elevated risk of various cancer types, including but not limited to breast, colorectal, endometrial, and kidney cancers [47]. These investigations not only quantify the increased risk but also delve into the temporal aspects, exploring how the duration and timing of obesity influence cancer development [48]. Moreover, epidemiological evidence contributes valuable insights into the impact of obesity on cancer outcomes, such as tumor aggressiveness, response to treatment, and overall survival rates [5,32]. The inclusion of this epidemiological perspective in our review enhances the comprehensiveness of our analysis, providing a broader contextual understanding of the public health implications and informing strategies for cancer prevention and intervention.

## 4. Cancer Rehabilitation

Cancer rehabilitation is a multidisciplinary approach to address the physical, emotional, and functional needs of individuals who have been diagnosed with cancer [49,50]. Cancer rehabilitation implies comprehensive cancer care, within a broader framework, that extends to and beyond cancer diagnosis and treatment, also considering survivorship. This approach recognizes that both cancer and its treatments can have an important impact on a person’s overall well-being [49,50]. In this context, cancer rehabilitation aims to enhance the quality of life, reduce symptoms burden and cancer side effects, and support individuals throughout their treatment [12]. The cornerstone of cancer rehabilitation is the multidisciplinary and interdisciplinary care involving a team of healthcare professionals, including physicians, surgeons, nurses, physical therapists, occupational therapists, speech therapists, dietitians, and psychologists, all working together to address the multicomponent disability characterizing cancer patients [51,52,53].

In this scenario, physical therapy is a key component of a comprehensive rehabilitation approach in cancer patients. In particular, cancer and its treatments can lead to physical impairments, such as muscle weakness, joint stiffness, loss of balance, and fatigue [12,54,55].

The study by Elme et al. [56] reported that obese patients might be characterized by a higher physical impairment compared to non-obese ones. Moreover, the authors highlighted that the strict connection between obesity, sedentary behavior, reduced physical performance, and higher cardiovascular risk might have detrimental consequences in HR-QOL [56]. As a result, a specific therapeutic intervention targeting physical performance impairment is needed, especially in obese patients due to its negative impact on overall wellbeing.

In addition, cancer patients might frequently be characterized by sarcopenia that can be secondary to chronic inflammation mediated by cytokines (associated with health conditions, including tumors), negatively impacting the outcomes of patients with cancer [57]. Studies have found that sarcopenia is often accompanied by an increase in the quantity of adipose tissue [58,59]. Consequently, the concept of sarcopenic obesity has emerged, representing a combination of sarcopenia and obesity [60]. In this context, physical therapists assess and work with patients to improve mobility, strength, and functional abilities. They may use exercises, manual techniques, and assistive devices to enhance physical function [12,54,55,61,62,63]. On the other hand, several barriers might affect an obese-specific approach since limb weight burden, limitations in joint mobility due to adipose tissue, and other anatomical constraints pose distinctive challenges [64]. The excessive weight carried by limbs complicates exercises and mobility, while adipose tissue can restrict joint movements, impacting the range of motion and complicating specific rehabilitation interventions [65]. These barriers underline the role of a specialized approach, considering the physical limitations imposed by obesity to ensure effective rehabilitation strategies addressing the multicomponent disability characterizing BC survivors with obesity.

Cancer-related fatigue (CRF) is a disabling issue, frequently complained by individuals undergoing cancer treatment [66]. While its pathogenesis has never been fully characterized, it has been reported that obesity might have a key role in fatigue development. In particular, the study by Inglis et al. [67] reported that obese patients have a higher level of fatigue compared to normal-weight patients. The mechanisms related to this linking might include systemic inflammation, mitochondrial dysfunction, altered oxidative stress response, impaired immune system function, and changes in energy metabolisms [12,17].

Rehabilitation programs provide techniques to counteract fatigue, including energy conservation strategies, graded exercise, and lifestyle modifications [68,69]. Occupational therapists assist patients in regaining independence in daily activities, such as bathing, dressing, and cooking. They may provide adaptive equipment and teach strategies to overcome physical limitations [51].

A common concern for patients is cancer-related pain [70]. Cancer rehabilitation includes strategies for pain management, which may involve medications, physical therapy, and interventions such as nerve blocks or acupuncture to alleviate pain and improve comfort [71,72]. In the pain management field of breast cancer survivors, growing attention is rising on joint-related symptoms such as pain, stiffness, and arthralgia are commonly linked to antihormonal treatment. Interestingly, the incidence of arthralgia is reported to be higher in obese patients, probably due to the higher doses of sex hormones secreted by the adipose tissue in obese patients [73]. In these people, it has been reported that greater sexual hormonal depletion related to antihormonal treatment in obese patients could have a role in improving pain related to arthralgia [74].

On the other hand, cancer treatments, such as surgery or radiation, can lead to lymphatic system pathologies such as axillary web syndrome and lymphedema [75,76]. Rehabilitation therapists can offer manual lymphatic drainage, compression therapy, and exercises to manage lymphedema [17,75,77,78]. In obese patients, a higher incidence of lymphedema has been reported with several studies underlining that obesity should be considered one of the most important patient characteristics to be considered in a precise risk assessment [79]. Moreover, obese patients might be characterized by reduced upper limb function that might be further impaired by lymphedema development with detrimental consequences in reaching and grasping function [80,81].

Moreover, for individuals with head and neck cancers, rehabilitation may encompass speech and swallowing therapy to address difficulties with speech and eating. Therapists work on improving communication and safe swallowing techniques [82]. In addition, maintaining proper nutrition is essential during cancer treatment since it reduces the burden of symptoms, reduces side effects, and reduces the physical and psychological consequences of cancer treatments [83]. In cancer rehabilitation, personalizing treatment in obese patients is crucial to improve functional outcomes, providing tailored nutritional counseling and support to address their specific dietary requirements, manage potential weight-related complications, and enhance their nutritional status as part of their comprehensive cancer care plan [84].

Lastly, cancer’s impact on mental health is widely noted, with growing reports underlining the role of cancer rehabilitation in managing not only physical issues but also improving psychological and emotional well-being [85,86,87]. In this context, the concept of prehabilitation has been recently proposed to encompass a proactive approach aimed at preparing individuals who are diagnosed with cancer for the physical and mental challenges they may face during their cancer treatments [88]. It involves a comprehensive assessment of a patient’s overall health, including their physical fitness, nutritional status, and emotional well-being, before they begin cancer therapies such as surgery, chemotherapy, or radiation [88]. The primary goal of prehabilitation is to enhance the patient’s physical and psychological resilience, thereby potentially reducing the impact of treatment-related side effects, accelerating recovery, and ultimately improving the overall success of their cancer treatment [89]. Prehabilitation strategies may include personalized exercise programs, nutritional guidance, stress management techniques, and psychological support, all designed to optimize a patient’s health and well-being as they embark on their cancer care pathway [89].

In this context, research is now emphasizing the potential role of specific pharmacological management of obese patients [90,91]. In recent years, pharmacological molecules including orlistat, liraglutide, bupropion/naltrexone, cathin, phentermine/topiramate, and lorcaserin were integrated into the management of obesity and were pharmacological treatments approved by the FDA [90,91]. While growing interest has been reported in the pharmacological management of obese patients, it should be noted that a lifestyle approach combined with a specific nutritional intervention still remains the key component of obesity treatments. Thus, it is not surprising that evidence supports the integration of these pharmacological approaches combined with specific exercise therapies [92].

On the other hand, it has been reported that drugs regulating metabolisms might have a protective role in cancer development [93]. In this context, these drugs may modulate pivotal signaling pathways implicated in cancer progression, including PI3K/AKT, MAPK, and AMP-activated protein kinase (AMPK) [94]. Furthermore, potential impacts on the tumor microenvironment, and inflammatory responses are explored, providing a more nuanced understanding of the advantages and disadvantages tied to the integration of anti-obesity drugs into the rehabilitation paradigm [95,96].

As a result, the potential synergisms between anti-obesity medications and rehabilitation strategies might be considered in the multidisciplinary management of obese patients with breast cancer. The dynamic interplay between pharmacological and rehabilitation-induced adaptations at the cellular and systemic levels might provide synergistic therapeutic effects [17,63]. However, evidence in this specific context is still lacking, and further studies should clarify the role of anti-obesity drugs in the rehabilitation management of obese patients with BC.

Cancer rehabilitation should be closely individualized on cancer characteristics, chancer treatments and patients’ characteristics, including obesity, since it impacts on cancer complication development and functional consequences. Cancer rehabilitation aims to empower cancer survivors care, to regain control of their lives, minimize the impact of cancer and its treatments, and ultimately improve their overall quality of life. This holistic approach plays a vital role in the comprehensive care and support of individuals affected by or at risk of cancer, promoting physical and emotional well-being from prevention, diagnosis through survivorship.

## 5. Role of Cancer Rehabilitation in Managing Obesity-Related Breast Cancer: A Practical Approach

There is a well-established relation between obesity and an increased risk of developing postmenopausal BC. Additionally, being overweight or obese as a BC patient implicates elevated risk of developing comorbidities, poorer surgical outcomes, a higher risk of lymphedema, an increased risk of fatigue, functional decline, and poorer health and overall quality of life and a general poorer overall and breast cancer-specific survival, with higher mortality rates [97,98]. Moreover, pain control could be less optimal due to the pharmacodynamic changes in obesity [99]. Table 2 summarizes a practical approach to rehabilitative needs in these patients. The risks and implications of obesity in BC highlight the importance of weight management in BC patients to potentially avoid adverse sequelae and late effects, as well as to improve overall health and possibly survival [97,98]. Intentional weight loss plays an important role in rehabilitation and recovery for BC patients, involving lifestyle interventions (dietary and behavior modification, increased aerobic and strength training exercise) that could offer a promising strategy to reverse the cancer-promoting effects of obesity [27,97,98].

According to the “Guidelines on Physical and Rehabilitation Medicine professional practice for adults with obesity and related comorbidities” from the International Society of Physical and Rehabilitation Medicine and the European Society of Physical and Rehabilitation Medicine [100], there is strong recommendation for prescribing at least 150 min per week of moderate-intensity exercise. The recommendation is based on the association of increased exercise duration with more marked reduction in weight and BMI compared to lower frequency exercise programs [100]. This duration falls within the WHO recommended amount of physical activity for adults living with chronic conditions; the document also states that engaging in physical activity is deemed safe for adults living with cancer, provided there are no contraindications, though, in general, the advantages surpass the potential risks [101]. Moreover, a recent systematic review has proven that exercise interventions in BC patients determine a biochemical modification at the level of metabolic and inflammatory biomarkers [17], which could be beneficial in tackling obesity-related dysfunctions. However, it is important to underline that the most beneficial effect on weight loss comes from the combination of physical activity and diet, rather than one or the other by itself [102].

A recent systematic review of oncological guidelines for rehabilitation indication provides specific rehabilitation and exercise recommendations for BC patients [103]. Rehabilitative interventions are indicated for common impairments related to breast cancer, such as fatigue, cognitive deficits, pain, neuropathy, lymphedema, musculoskeletal impairment, and sexual dysfunction [12,103]. In this context, several guidelines recommended referral for rehabilitation based on treatment timing and symptom onset to address issues such as upper extremity exercises after surgery, lymphedema management, and sexual and hormone-related symptoms [75,77,103,104,105].

In the context of cancer-related fatigue, exercise might represent a pivotal element aiming at both alleviating fatigue and contributing to weight management. Engaging in regular physical activity not only combats the debilitating effects of cancer-related fatigue but also plays a crucial role in reducing excess weight and inflammation in obese individuals [12]. Studies have indicated that exercise induces biochemical modifications, influencing metabolic and inflammatory biomarkers, which can be particularly advantageous in addressing dysfunctions associated with obesity [17].

In addition, lymphedema might be considered a distinctive challenge in obese individuals due to associated morphological changes, posing complexities in its management within the context of rehabilitation [16,75,77,79]. In this context, digital advances might offer innovative tools including three-dimensional assessment of lymphedema, thereby overcoming barriers for a precise assessment of these patients [106,107]. Moreover, the personalized approach to rehabilitation becomes imperative, necessitating tailored interventions that acknowledge and adapt to altered morphologies observed in obese patients experiencing lymphedema [108]. Integrating self-adjusting braces and personalized rehabilitation regimens emerges as a promising strategy to address the unique anatomical changes and mitigate the impact of lymphedema [109]. These approaches not only accommodate the altered morphologies encountered in obese individuals but also offer a more personalized and adaptive means to manage lymphedema effectively [16]

On the other hand, the burden of excess weight in obese individuals often leads to structural damage in joint components, posing significant challenges in conventional rehabilitative approaches [110]. The increased load on weight-bearing joints can exacerbate joint-related issues, complicating the effectiveness of traditional rehabilitation methods [111]. To address these challenges, rehabilitation strategies employing aquatic therapy present a promising therapeutic approach alleviating weight-bearing stress on joints, enabling individuals to engage in exercises and movements with reduced impact [112]. Water-based rehabilitation strategies offer a supportive medium for obese individuals, facilitating improved mobility and function without subjecting the joints to excessive stress [113]. Incorporating aquatic therapy into rehabilitation programs tailored for obese individuals presents an innovative approach to mitigate the limitations posed by excess weight, offering a more effective and comfortable avenue for rehabilitation.

Despite these considerations, the occurrence of small joint pain in patients undergoing treatment with aromatase inhibitors poses a significant challenge, particularly in obese individuals [114]. However, exercise strategies coupled with the utilization of vibrating platforms have garnered substantial evidence in addressing this concern [71]. For obese patients experiencing small joint pain, the implementation of exercise regimes tailored to their needs, emphasizing low-impact activities and joint-specific exercises, has shown promise in managing discomfort while enhancing joint mobility and function [115]. Furthermore, the application of vibrating platforms in obese individuals has exhibited considerable evidence in improving musculoskeletal health. The vibrational stimulus delivered through these platforms has demonstrated benefits in enhancing bone density and joint proprioception, potentially alleviating small joint pain and discomfort experienced by patients on aromatase inhibitors [63,115].

Figure 2 summarizes the tailored rehabilitative interventions addressing specific challenges in obese BC patients.

In conclusion, different rehabilitation strategies should be considered in obese patients with BC. In this context, a precise assessment of the multifaced disability characterizing disability in BC survivors with obesity should be the cornerstone for a precise rehabilitation approach. Despite these considerations, few studies are currently available about this topic, and further research is needed to support an evidence-based approach in cancer patients with obesity.

## 6. Challenges, Future Directions, and Study Limitations

Despite the above-discussed evidence of the benefit of including interdisciplinary management of cancer rehabilitation, there are still several barriers that need to be tackled to implement programs efficiently.

First of all, establishing an extensive collaborative network can be challenging, as it requires organizational changes and a commitment to teamwork [116]. Then, a prominent challenge lies in the awareness gap surrounding cancer rehabilitation. This lack of awareness prevails among patients who may not be familiar with the existence and potential benefits of these services. Similarly, healthcare providers and institutions often remain unaware of the vital role that cancer rehabilitation plays [78,88,117]. Moreover, resource allocation presents another limit. Cancer rehabilitation programs require specialized personnel, equipment, and dedicated facilities. However, the financial constraints faced by healthcare institutions can severely impede the expansion of these services [88,118,119].

To the previously mentioned barriers, obesity-treatment specific barriers must be added. From the patient perspective, obesity is often not recognized as a chronic and relapsing disease, the cost of treatment is high, and certain comorbidities and medications can impose limitations on physical activity. Additionally, patients may lack a strong support network of family, friends, or peers, which is necessary for successful adherence to long-term lifestyle changes [120]. Some physician factors can act as barriers as well to obesity management, including a lack of time during consultations and insufficient training and counseling skills [120].

Addressing these challenges demands a combined effort at multiple levels, involving healthcare systems, healthcare providers, and individual patients. To overcome the awareness gap, divulgation efforts must be intensified both within the healthcare sector and at community levels. Healthcare professionals should be educated on the importance of early referral to rehabilitation services for individuals facing obesity-related cancer. Resource allocation must invest in specialized training for rehabilitation professionals, establishing dedicated rehabilitation units, and ensuring equitable access to these services as a core component of cancer care, while exploring the economic implications and their potential to reduce overall healthcare expenditure. Furthermore, the adoption of integrated care models is crucial to facilitate seamless interdisciplinary collaboration.

These models promote the holistic integration of cancer rehabilitation into the broader cancer care continuum, and could enhance care coordination and, subsequently, patient outcomes.

Personalized rehabilitation approaches could have the potential to optimize the effectiveness of rehabilitation programs, tailoring interventions to the unique characteristics and needs of individual patients. Telehealth technologies offer novel possibilities for delivering cancer rehabilitation services, particularly to individuals with limited access [121]. Research into the feasibility and efficacy of telerehabilitation for obesity-related cancer is an area that merits exploration.

Artificial intelligence (AI) has shown promising potential in improving cancer treatment for patients, offering more personalized and effective care [122,123]. AI algorithms can analyze a patient’s medical history, genetic data, and lifestyle factors, including obesity, to assess their risk of developing cancer, cancer recurrence, or cancer-treatment side effects [18,124,125]. This enables earlier detection and intervention. Artificial intelligence might assist oncologists in tailoring treatment plans based on the patient’s unique physiological and genetic characteristics, including their obesity condition avoiding complications in cancer treatment planning. This personalized approach could help optimize the effectiveness of therapies while minimizing potential side effects.

To improve our comprehension of the intricate interactions between obesity and breast cancer, research should explore new directions, with a focus on personalized approaches. Future investigations should prioritize the identification of novel biomarkers and genetic factors to enhance the risk stratification for obesity-related breast cancer. Exploring the impact of the gut microbiome and metabolomics on cancer progression holds promise for targeted interventions. Combining digital health technologies to provide real-time tracking, particularly telemedicine, could revolutionize comprehensive oncologic rehabilitation accessibility for obese cancer patients. Longitudinal studies tracking lifestyle interventions’ long-term effects and assessing patient-reported outcomes are essential. Additionally, integrating artificial intelligence to predict treatment responses and tailoring rehabilitation plans offers a frontier for optimizing outcomes. Such focused research directions aim to refine our understanding and ultimately improve the personalized management of obesity-related breast cancer.

Despite offering a comprehensive overview of the topic, this narrative review presents inherent limitations. First, objectivity can encounter potential selection bias and the risk of missing evidence. Moreover, the absence of statistical synthesis and a formal quality assessment may limit the robustness of these conclusions drawn from the literature. Lastly, the vastness of the topic treated may introduce subjectivity bias through interpretation of studies. However, even in the presence of these drawbacks, this work can serve as initial exploration, offering insights and identifying research gaps, and providing an initial practical approach when facing rehabilitation of obesity-related BC.

## 7. Conclusions

In conclusion, the findings from this comprehensive review underline the role of tailored rehabilitation strategies designed to address the disabling consequences faced by obese patients with cancer. In this context, emerging strategies including digital implementation, artificial intelligence, and tele-health approach hold great promise as tools to support the continuity and sustainability of a comprehensive rehabilitation intervention. Future research might further elucidate the pivotal role of comprehensive rehabilitation programs in enhancing improvements and sustained care for obese cancer patients, minimizing disabling consequences, and optimizing overall well-being in the long-term management of cancer patients.

## Figures and Tables

**Figure 1 cancers-16-00521-f001:**
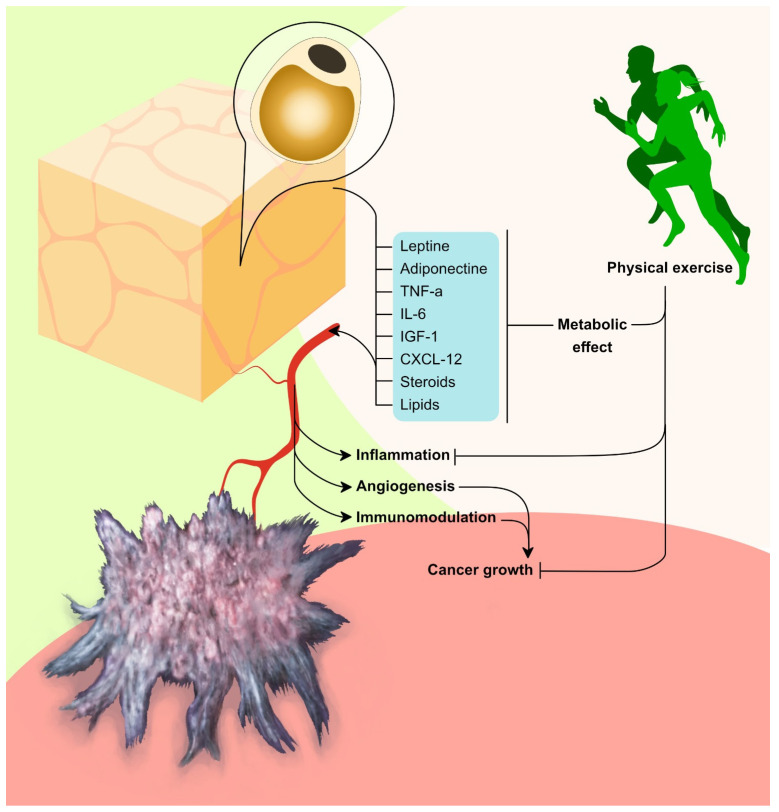
This figure summarizes the adipose tissue impact on tumor environment and the potential effects of physical exercise in obese patients with BC.

**Figure 2 cancers-16-00521-f002:**
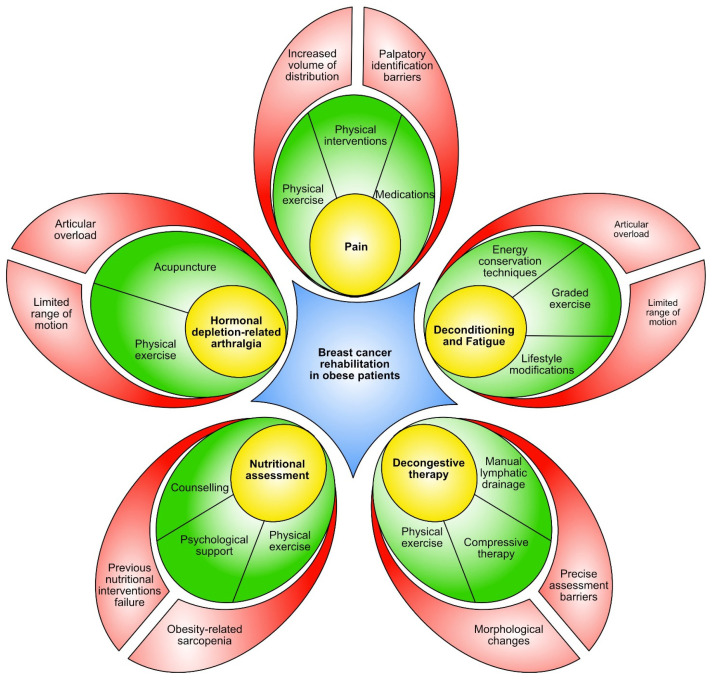
Rehabilitation barriers and tailored intervention strategies for obese breast cancer patients.

**Table 1 cancers-16-00521-t001:** Spider tool search strategy.

S	PI	D	E	R
Sample	Phenomenon of Interest	Design	Evaluation	Research Type
Obese patients with breast cancer	Rehabilitation Strategies	Qualitative research including reviews, case studies, observational studies, and focus groups. Quantitative and mixed-method research including clinical trials, randomized clinical trials, cohort studies, and cross-sectional studies. Peer-reviewed research.	Rehabilitation Outcomes	Qualitative
“Cancer”, “Breast”, “Obesity”	“Exercise”, “Nutrition”, “Rehabilitation”, “Physical Activity”, “Lifestyle”		“Function”, “Body Composition”, “Pain”, “Performance”, “Disability”, “Quality of Life”, “Cancer Survivorship”	

**Table 2 cancers-16-00521-t002:** A practical approach in rehabilitation managing obesity-related breast cancer.

Issue	Practical Approach
Weight Management	Intentional weight loss
	Lifestyle interventions (dietary modification, exercise, psychological support)
Physical Activity Recommendations	At least 150 min of moderate-intensity exercise per week
	More significant body mass index reduction with increased exercise duration
Cancer-Related Fatigue	Exercise as a pivotal element for fatigue alleviation
Cancer-Related Lymphedema	Precise assessment through innovative tools (e.g., three-dimensional assessment of lymphedema)
	Personalized self-adjusting braces and rehabilitation regimens
Surgery-Related Joint Dysfunction	Active and passive early mobilization, and upper extremity exercises after surgery
Weight-Related Joint Pain	Aquatic therapy to alleviate weight-bearing stress on joints
Small Joint Pain in Aromatase Inhibitor Treatment	Tailored exercise regimes emphasizing low-impact activities and joint-specific exercises
	Consideration of vibrating platforms to improve musculoskeletal health

## Data Availability

The data can be shared up on request.

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
