# Peer review of "Obesity and Cancer Rehabilitation for Functional Recovery and Quality of Life in Breast Cancer Survivors: A Comprehensive Review"

_cancers, 2024, doi:10.3390/cancers16030521_

Round 1

Reviewer 1 Report

Comments and Suggestions for Authors

I've read with attention the review by Lippi et al. that is potentially of interest, update and adequately referenced. The figures are very nice and add value to the review. I only would suggest to summarize eventual data coming from interventional study in a further table. A mention on the potential role of antiobesity drugs in the mangament of these patients (including the eventual risk of pharmacological interaction) should be added as well.

Author Response

Reviewer #1:

I've read with attention the review by Lippi et al. that is potentially of interest, update and adequately referenced. The figures are very nice and add value to the review.

Dear Reviewer,

Many thanks for your letter and kind comments concerning our manuscript. We are glad that the Reviewer has appreciated our work.

 I only would suggest to summarize eventual data coming from interventional study in a further table.

We would like to thank the Reviewer for the insightful comment. We improved the manuscript providing a table summarizing a possible interventional approach in accordance with the Reviewer’s comment. Thus, Table 2 was added to the manuscript.

A mention on the potential role of antiobesity drugs in the management of these patients (including the eventual risk of pharmacological interaction) should be added as well.

We would like to thank the Reviewer for the insightful comment. We improved the manuscript providing a paragraph about antiobesity drugs and its interactions, focusing on antioncogenic effects and potential synergisms with rehabilitation, in accordance with the Reviewer’s comment. The paragraph can be found between line 337 and line 360, as it follows.

“In this context, research is now emphasizing the potential role of specific pharmaco-logical management of obese patients [95, 96]. In recent years, pharmacological molecules including orlistat, liraglutide, bupropion/naltrexone, cathin, phentermine/ topiramate, and lorcaserin were integrated into the management of obesity and were pharmacological treatments approved by the FDA [95, 96]. While growing interest has been reported in the pharmacological management of obese patients, it should be noted that a lifestyle ap-proach combined with a specific nutritional intervention still remains the key component of obesity treatments. Thus, it is not surprising that evidence supports the integration of these pharmacological approaches combined with specific exercise therapies [97].

On the other hand, it has been reported that drugs regulating metabolisms might have a protective role in cancer development [98]. In this context, these drugs may modulate pivotal signaling pathways implicated in cancer progression, including PI3K/AKT, MAPK, and AMP-activated protein kinase (AMPK) [99]. Furthermore, potential impacts on the tumor microenvironment, and inflammatory responses are explored, providing a more nuanced understanding of the advantages and disadvantages tied to the integration of anti-obesity drugs into the rehabilitation paradigm [100, 101].

As a result, the potential synergisms between anti-obesity medications and rehabilitation strategies might be considered in the multidisciplinary management of obese patients with breast cancer. The dynamic interplay between pharmacological and rehabilitation-induced adaptations at the cellular and systemic levels might provide synergistic therapeutic effects [17, 66]. However, evidence in this specific context is still lacking and further studies should clarify the role of anti-obesity drugs in the rehabilitation management of obese patients with BC.”

Reviewer 2 Report

Comments and Suggestions for Authors

Dear authors,

Your review is both interesting and important, but there is room for improvement.

  1. - The review is not free from typos. An example sentence for improvement could be: 'In light of these considerations, the aim of this narrative review is to summarize the current evidence about the link between obesity, cancer, and rehabilitation to provide deeper insight into the most effective rehabilitation interventions for obese individuals with breast cancer.'

  2.  
  3. - Concerns arise from the SPIDER tool search strategy outlined in Table 1. Firstly, specify the study designs considered, instead of using the term 'any,' as there are designs you may not have included. Also, clarify in the table that the reliance on qualitative research pertains to the type of data synthesis for the review, not the types of studies considered.

  4.  
  5. - While the authors discuss in detail various biological pathways linking obesity and cancer, they overlook the existing epidemiological evidence on the topic. It would be advisable to allocate a subsection (or a separate section within the review) to discuss findings from epidemiological studies. This is particularly relevant given the authors' statement that they considered all types of designs in preparing this review.

  6.  
  7. - The subsection 'Challenges and Future Directions' largely focuses on practical aspects, but it would be desirable to also consider future directions for research.

Author Response

Reviewer #2:
Dear authors,

Your review is both interesting and important, but there is room for improvement.

Dear Reviewer,

Many thanks for your letter and kind comments concerning our manuscript. We are glad that the Reviewer has appreciated our work.

- The review is not free from typos. An example sentence for improvement could be: 'In light of these considerations, the aim of this narrative review is to summarize the current evidence about the link between obesity, cancer, and rehabilitation to provide deeper insight into the most effective rehabilitation interventions for obese individuals with breast cancer.'

We would like to thank the Reviewer for the insightful comment. We have improved the sentence in accordance with the Reviewer’s comment. Moreover, an extensive English revision has been performed in order to meet the Reviewer’s quality requirements. The paragraph can be found between line 92 and line 95, as it follows.

“In light of these considerations, the goal of this narrative review is to outline the cur-rent evidence regarding the association between obesity, cancer, and rehabilitation. The aim is to provide a more in-depth understanding and insight into the most effective rehabilitation intervention for individuals with both obesity and breast cancer.”.

- Concerns arise from the SPIDER tool search strategy outlined in Table 1. Firstly, specify the study designs considered, instead of using the term 'any,' as there are designs you may not have included. Also, clarify in the table that the reliance on qualitative research pertains to the type of data synthesis for the review, not the types of studies considered.

We would like to thank the Reviewer for the insightful comment. We improved the Table 1 providing a more precise description of the inclusion criteria for study designs. In addition, we improved the Materials and Methods section, by clarifying the method used to perform a qualitative analysis.

The paragraph can be found between line 121 and line 125, as it follows.

“Given the heterogeneity of the included studies and the narrative review design, a qualitative synthesis approach was employed, presenting all outcome data in a narrative format. The reliance on qualitative research pertains to the type of data synthesis for the review, not the types of studies considered.”.

- While the authors discuss in detail various biological pathways linking obesity and cancer, they overlook the existing epidemiological evidence on the topic. It would be advisable to allocate a subsection (or a separate section within the review) to discuss findings from epidemiological studies. This is particularly relevant given the authors' statement that they considered all types of designs in preparing this review.

We would like to thank the Reviewer for the insightful comment. We improved the manuscript including a new paragraph in “Biological basis linking Obesity and Cancer” Section  to better characterize the evidence provided by epidemiological studies in accordance with the Reviewer’s comment. The paragraph can be found between line 227 and line 238, as it follows.

“Epidemiological evidence plays another pivotal role in clarifying the intricate relationship between obesity and cancer at the population level [49]. Numerous epidemiological studies have consistently demonstrated associations between excess body weight and an elevated risk of various cancer types, including but not limited to breast, colorectal, endometrial, and kidney cancers [50]. These investigations not only quantify the increased risk but also delve into the temporal aspects, exploring how the duration and timing of obesity influence cancer development [51]. Moreover, epidemiological evidence contributes valuable insights into the impact of obesity on cancer outcomes, such as tumor aggressiveness, response to treatment, and overall survival rates [6, 34]. The inclusion of this epidemiological perspective in our review enhances the comprehensiveness of our analysis, providing a broader contextual understanding of the public health implications and in-forming strategies for cancer prevention and intervention.”.

- The subsection 'Challenges and Future Directions' largely focuses on practical aspects, but it would be desirable to also consider future directions for research.

We would like to thank the Reviewer for the insightful comment. We improved this Section by characterizing potential-research future direction in accordance with the Reviewer’s comment. We improved the manuscript including different paragraph in “Challenges, Future Directions, and Study Limitations” Section. The paragraph can be found between line 516 and line 528, as it follows.

“To improve our comprehension of the intricate interactions between obesity and breast cancer, research should explore new directions, with a focus on personalized approaches. Future investigations should prioritize the identification of novel biomarkers and genetic factors to enhance risk stratification for obesity-related breast cancer. Exploring the impact of the gut microbiome and metabolomics on cancer progression holds promise for targeted interventions. Combining digital health technologies to provide re-al-time tracking, particularly telemedicine, could revolutionize comprehensive oncologic rehabilitation accessibility for obese cancer patients. Longitudinal studies tracking life-style interventions' long-term effects and assessing patient-reported outcomes are essential. Additionally, integrating artificial intelligence to predict treatment responses and tailoring rehabilitation plans offers a frontier for optimizing outcomes. Such focused research directions aim to refine our understanding and ultimately improve the personalized management of obesity-related breast cancer.”.

Reviewer 3 Report

Comments and Suggestions for Authors

Dear authors, 

First of all I would like to congratulate you on the research presented. It is much needed in your field of research. Regarding the quality of the study, it presents a high methodological quality. The necessary criteria for a literature review are followed. 

As suggestions for improvement I propose: 

Put keywords that are not found in the title. 

Replace quotations from the discussion and introduction prior to 2019. This is proposed mainly to improve the contextualisation of the research. 

Point out the limitations of your research. This is vital and necessary for any research. Section 6 should be entitled limitations and applicability of this study. Likewise, point out the applications that your research presents to the field of study. 

Comments on the Quality of English Language

English level is fine.

Author Response

Reviewer #3:

Dear authors,

First of all I would like to congratulate you on the research presented. It is much needed in your field of research. Regarding the quality of the study, it presents a high methodological quality. The necessary criteria for a literature review are followed.

Dear Reviewer,

Many thanks for your letter and kind comments concerning our manuscript. We are glad that the Reviewer’s has appreciated our work.

Put keywords that are not found in the title.

We would like to thank the Reviewer for the insightful comment. We improved the keywords in accordance with the Reviewer’s comment. The keywords can be found between line 43 and line 45, as it follows.

“Breast Cancer; Morbid Obesity; Obesity Management; Rehabilitation; Personalized Medicine; Physical Exercise; Health-Related Quality Of Life; Treatment Outcome”

Replace quotations from the discussion and introduction prior to 2019. This is proposed mainly to improve the contextualisation of the research.

We would like to thank the Reviewer for the insightful comment. We carefully reviewed our references and updated the improper ones in accordance with the Reviewer’s comment.

Point out the limitations of your research. This is vital and necessary for any research. Section 6 should be entitled limitations and applicability of this study. Likewise, point out the applications that your research presents to the field of study.

We would like to thank the Reviewer for the insightful comment. We improved the title of the chapter to “Challenges, Future Directions, and Study Limitations” Section in accordance with the Reviewer’s comment. Moreover, we added a limitations and application paragraph between line 529 and line 537, as it follows.

“Despite offering a comprehensive overview of the topic, this narrative review presents inherent limitations. First of all, objectivity can encounter potential selection bias and the risk of missing evidence. Moreover, the absence of statistical synthesis and a formal quality assessment may limit the robustness of these conclusions drawn from the literature. Lastly, the vastness of the topic treated may introduce subjectivity bias through interpretation of studies. However, even in the presence of these drawbacks, this work can serve as initial exploration, offering insights and identifying research gaps, and providing an initial practical approach when facing rehabilitation of obesity-related BC”.

Round 2

Reviewer 2 Report

Comments and Suggestions for Authors

Well done!